# Application of Stable Isotope Analysis for Detecting the Geographical Origin of the Greek Currants “Vostizza”: A Preliminary Study

**DOI:** 10.3390/foods12081672

**Published:** 2023-04-17

**Authors:** Anna-Akrivi Thomatou, Eleni C. Mazarakioti, Anastasios Zotos, Achilleas Kontogeorgos, Angelos Patakas, Athanasios Ladavos

**Affiliations:** 1Department of Food Science & Technology, University of Patras, 30100 Agrinio, Greece; 2Department of Biosystems Science and Agricultural Engineering, University of Patras, 30200 Messolongi, Greece; 3Department of Agriculture, International Hellenic University, 57001 Thessaloniki, Greece

**Keywords:** geographical origin, Greek currants, isotope-ratio mass spectrometry (IRMS)

## Abstract

There is a plethora of food products with geographical indications registered in the European Union without any study about their discrimination from other similar products. This is also the case for Greek currants. This paper aims to analyze if stable isotope analysis of C, N, and S could discriminate the Greek currants “Vositzza”, registered as a product of Protected Designation of Origin, from two other currants registered as products of Protected Geographical Indication coming from neighboring areas. The first results show that the stable isotope ratio of sulfur is not detectable due to the very low sulfur content in the samples, and the analysis should be based on the stable isotope ratios of carbon and nitrogen to discriminate these products. The mean value of δ^15^N (1.38‰) of PDO “Vostizza” currants is lower than that of currants grown outside the PDO zone (2.01‰), while the mean value of δ^13^C of PDO “Vostizza” currants is higher (−23.93‰) in comparison to that of currants grown outside the PDO zone (−24.83‰). Nevertheless, the results indicate that with only two isotopic ratios, discrimination could not be achieved, and further analysis is required.

## 1. Introduction

Food products with geographical indications increasingly attract consumers’ interest, along with their willingness to pay more for such products and their distinctive characteristics. At the same time, the higher prices of these products motivate mislabeling or even adulteration, as an easy way for more profits, at least in the short run [1,2]. Food traceability, which has become the basis for food safety police, has led the EU to introduce the 178/2002/EC Traceability Regulation that defines “food and feed traceability” and can give information about the origin of products through the food chain [3]. Food authenticity requirements have forced the EU to recognize the importance of geographical origin of food. Food authenticity refers to the extent to which a food product is what it claims to be. It involves ensuring that the food product is not adulterated or mislabeled, and that it meets certain quality standards or regulations. 

In other words, it is the assurance that a food product is genuine, has not been tampered with, and is safe for consumption. In 1992, the EU enacted the Protected Designation of Origin (PDO) and Protected Geographical Indication (PGI) schemes (Regulation 2081/92/EEC), followed by Regulation 510/2006/EEC and amended by Regulation 1151/2012/EU, which are European laws against mislabeling [1,4]. 

Certainly, verifying food authenticity is an important process for a number of reasons, including the safety of consumers, consumer confidence, fair trade, and regulatory compliance. Firstly, adulterated or mislabeled foods can be a serious health risk to consumers. Verifying the authenticity of food products could help mitigate risks about safety by ensuring that the food products are safe for consumption and free from harmful contaminants. This is particularly important for those with food allergies or other dietary restrictions. Secondly, food authenticity verification provides consumers with confidence that the food they are buying is what it claims to be. This helps build trust between consumers and the food industry and is especially important at a time when food fraud is becoming more prevalent. Thirdly, verifying food authenticity helps ensure that food producers receive fair compensation for their products. It also helps ensure that consumers are not deceived or misled about the origin or quality of the food they are purchasing. This promotes fair trade practices and helps to support the livelihoods of farmers and food producers. Lastly, many countries have regulations in place that require food producers to meet certain quality and labeling standards. Verifying food authenticity helps ensure that producers are in compliance with these regulations, which can have legal and financial implications for businesses. 

Geographical origin is an essential part of food authenticity since there is a direct link between food characteristics, such as flavor, aroma, texture, and nutritional content, and the unique environmental and cultural factors of the production area. Geographical origin verification is particularly important for certain food products that are associated with specific regions or traditions, such as the PDO and the PGI products established in Europe. Verifying the authenticity of a PDO or a PGI product is important for promoting consumer safety and confidence, meeting regulatory standards, preserving cultural heritage, and supporting the local economy. Verifying the geographical origin of a food product involves analyzing various chemical, physical, and biological characteristics of the food.

Various analytical techniques, including chromatographic techniques, such as gas chromatography–mass spectrometry (GC-MS) [5] or high-performance liquid chromatography (HPLC) [6], nuclear magnetic resonance (NMR) [7], ultra-violet visible spectroscopy (UV-Vis) [8], near-infrared spectroscopy (NIR) [9], inductively coupled plasma mass spectrometry (ICP-MS) [10], DNA-based techniques [11], and isotope-ratio mass spectrometry (IRMS), have been proposed for the accurate determination of food geographical origin [12]. The most widely used technique for information about geographical origin is isotope-ratio mass spectrometry (IRMS), which has been used to determine the geographical origin of various agricultural products [12,13,14,15,16,17]. Stable isotope analysis relies on the measurement of the isotope ratio of elements and the fact that the ratio of these elements depends on the carbon fixation process for carbon isotope; soil nutrition for nitrogen isotope; climatic conditions, latitude, and elevation for hydrogen and oxygen isotopes; and distance from the sea and soil structure for sulfur isotopes [18]. 

The aim of this study was the differentiation of the geographical origin of “Vostizza” currants from the region of Egialia by means of stable isotope analysis and the use of EA-IRMS. As there are no studies about the stable isotope composition of “Vostizza” currants, these are the first published data. 

### The Examined Product

Worldwide, a very popular and beneficial vine product for human health is dried grapes, which are very rich in nutrients. The chemical composition of dried grapes is complex and includes a variety of bioactive compounds that contribute to their nutritional and health benefits. These compounds make dried grapes a nutritious addition to a healthy diet. Some examples of the key components of dried grapes’ chemical composition include antioxidants and a high percentage of anthocyanins, flavonoids, vitamin C, potassium, and fiber [19,20,21,22,23,24,25,26,27].

Dried grapes in Greece are usually known as raisins (dried white grapes) and currants (dried red grapes), mainly coming from different types of the Corinthian variety [7]. More specific currants are particularly high in anthocyanins, which are pigments that give the fruit its dark color. These compounds are powerful antioxidants that have been linked to a range of health benefits, including reduced inflammation. Currants are also rich in flavonoids, which are another type of antioxidant. These compounds are thought to play a role in protection against chronic diseases, such as cardiovascular diseases. Currants are a good source of vitamin C, which is an essential nutrient that has antioxidant properties and is important for immune function. In addition, currants are a good source of dietary fiber, which can help promote digestive health and lower the risk of chronic diseases. Finally, currants are a good source of potassium, which is an important mineral that plays a role in regulating blood pressure and promoting heart health.

In Greece, and especially in the wider area of Peloponnese, there are two types of Corinthian currants that hold a protected geographical indication scheme: Corinthian currants cultivated in the Egialia area and registered with the name “Vostizza” (PDO reg No. 442597/93) since 1993, and Corinthian currants cultivated in the Prefecture of Ilia registered as a PGI product under the name “Stafida Ilias” (PGI reg No. C233/2010) since 2010. There is also a PDO product based on the Corinthian variety that is cultivated on the island of Zakynthos, belonging to the Ionian Island Prefecture [28]. This product is registered with the name “Stafida Zakynthoy” (PDO reg No. C179/2007). Products that are registered as PDO or PGI must be grown and processed in a specific region of Greece to be considered authentic. This can help ensure that consumers are purchasing genuine “Vostizza” currants along with all its healthy compounds.

However, according to the Greek Ministry of Agriculture, “Vostizza” currants are the ones with the largest production volume, and their cultivated area reaches almost five thousand hectares, which is almost twice the cultivation area of “Stafida Ilias” currants and three times the area of “Stafida Zakynthoy” currants. Furthermore, “Vostizza” currants are one of the very first Greek products registered under the Protected Designation of Origin policy scheme. According to Vasilopoulou and Trichopoulou [23] “the world-renowned” currant under the name “Vostizza” is exclusively cultivated on the semi-mountainous and mountainous areas of the ancient town of Aeghio (its medieval name is Vostizza) from the 12th century. What differentiates “Vostizza” currants from other varieties is the slope of the ground of the vineyards and the microclimate of this region. Compared to other Corinthian currants, “Vostizza” currants present a higher total anthocyanin content and a lower total phenolics content and antiradical activity. “Vostizza” currants have unique physical characteristics, such as their small size, dark color, and wrinkled appearance. These characteristics can help distinguish them from other currants. The microclimate of the region differentiates “Vostizza” currants from currants produced in other regions and accounts for their exceptionally high reputation [23]. For these reasons, this study will focus mainly on “Vostizza” currants.

Verifying the authenticity of “Vostizza” currants could provide consumers with confidence that they are buying a genuine product and all its known and desired characteristics. This could help establish trust between consumers and the food industry that trade “Vostizza” currants, which hold a PDO. Increased trust consequently leads to increased sales for verified and authentic products. This has a direct impact on the prices of the products, especially for “Vostizza” currants, which constitute an important export for both Greece’s national agricultural economy and local economy, as well as for producers/farmers of the product, helping to support the livelihoods of farmers and other businesses involved in the production and sale of this product. At the same time, since “Vostizza” currant is a traditional Greek product that has been cultivated for centuries by small family-owned farms using traditional methods, it enhances the product’s unique characteristics, such as flavor and aroma. “Vostizza” currants are harvested by hand and then sun-dried, which further concentrates their flavor and sweetness. Thus, verifying their authenticity could help preserve this traditional way of cultivation, which is a cultural heritage, and ensure that this product remains true to its origins as it is described in the European Union’s PDO registration files. Additionally, this is a point of crucial importance since verifying the authenticity of “Vostizza” currants means that the product meets regulatory standards by ensuring that the product confronts specific quality and labeling requirements. 

Many studies have been carried out in order to determine the geographical origin of table grapes and wines using stable isotope analysis [29,30,31,32,33,34]. On the contrary, only a few studies have been performed for currants. Li et al. [35] determined the geographical origin of blackcurrants from different regions of China using stable isotopes. Perini et al. [36] used isotope-ratio mass spectrometry to verify the authentication of berries (strawberries, raspberries, blueberries, blackberries, and currants) collected from different countries. Dimitrakopoulou et al. [7] used the ^1^H-NMR non-targeted fingerprinting method for the differentiation of “Vostizza” currants from currants of other geographical origins (Zante, Kalamata, Nemea, and Amaliada), while Dimitrakopoulou et al. [11] discriminated “Vostizza” currants by using DNA-based methods. 

## 2. Materials and Methods

### 2.1. Sampling

In total, 100 currants samples were examined: eighty samples were from Egialia in Peloponnese in southern Greece (Figure 1), and the rest of the samples were from different areas neighboring Egialia. However, only Egialia produces Corinthian black currants “Vostizza” that have been registered since 1993 as a PDO product (file name PDO No. 1549/98), as already mentioned. The samples were labeled with numbers from “21–100” for the PDO samples and “1EP–20EP” for the rest of the samples (Figure 1). All samples came from different producers and were produced in 2021. 

### 2.2. Sample Preparation

Fifty grams of each currant sample were freeze-dried at −50 °C for 24 h, followed by homogenization in a mill (pulverisette 11, Fritsch GmbH Milling and Sizing). Then, the samples were divided into two sections for the determination of *δ*^15^N and δ^13^C. To determine *δ*^15^N, the homogenized sample was extracted in a mixture of acetone/distilled water (10% *v*/*v*) to remove sugars and lipids, as described in the ENV 13070 method, followed by centrifugation and picking up of the solid part, which was oven-dried at 50 °C for 2 h. The extraction was applied for the enrichment of nitrogen in the received solid. Afterward, it was further homogenized with a mortar. In order to determine δ^13^C, the homogenized sample was also oven-dried at 50 °C for 2 h and crushed with a mortar. After processing, each part was put in Eppendorf tubes and placed in a freezer until the IRMS analysis.

### 2.3. EA-IRMS Analysis

The isotopic analyses of carbon, nitrogen, and sulfur were performed by an Elementar Isoprime 100 Isotope-Ratio Mass Spectrometry (IRMS) instrument (IsoPrime Ltd., the village Cheadle Hulme, UK) coupled to an Elemental Analyzer (Elementar Vario Isotope EL Cube, Elementar Analysensysteme GmbH, Hanau, Germany). The samples, ≈1–2 mg, were weighed into tin capsules for measurement and were loaded on the auto-sampler of the IRMS analyzer. The results of the isotope ratio analyses were expressed as delta values *δ* (‰) and calculated according to the following equation:*δ* Χ (‰) = [(R_sample_/R_standard_) − 1] × 1000 (1)
where X is the isotope being studied (e.g., ^13^C, ^15^N, and ^34^S), R_sample_ is the isotope ratio of the measured element in its physical form (e.g., ^13^C/^12^C, ^15^N/^14^N, and ^34^S/^32^S) in the sample, and R_standard_ is the isotope ratio of the reference material.

Before sample analysis, instrument calibration was performed with the means of the reference materials with a known composition (standard substances). The standard substances were selected based on the isotope (C, N, and S) that was required for the analysis. In addition, physicochemical properties are an important factor for the selection of the reference materials, i.e., their isotope ratio should be near the one of the samples that was examined. 

The batch routine began with the analysis of a reference material, followed by a number of the samples, and ended with another reference material. In this study, IAEA-600 (Caffeine, δ^13^C_V-PDB_ = −27.77‰) and B259 (Sorgum flour IRMS Standard, with δ^13^C_V-PDB_ = −13.78‰) were the reference materials used for *δ*^13^C analysis. IAEA-N1 (Ammonium Sulfate, δ^15^N_Air_ = 0.43‰) and B259 (Sorgum flour IRMS Standard, with δ^15^N_Air_ = 1.58‰) were the reference materials used for δ^15^N analysis. IAEA-S1 (Silver Sulfide, δ^34^S = −0.3‰) and B259 (Sorgum flour IRMS Standard, with δ^34^S = 10.11‰) were the reference materials used for δ^34^S analysis. During sample analysis, a quality-control-check sample with B2155, (protein IRMS Standard, with δ^13^C_V-PDB_ = −26.98‰, δ^15^N_Air_ = 5.83‰, and δ^34^S = 6.18‰) was analyzed to test the results of our samples. 

## 3. Results and Discussion

Eighty samples from the cultivation zone of the PDO product “Vostizza”, along with twenty more samples coming from other areas outside the PDO zone of “Vostizza”, were analyzed. For each sample, two repetitions were performed. The mean values of δ^15^N_AIR_ (‰) and δ^13^C_V-PDB_ (‰) for the examined areas are presented in Table 1. Appendix A presents the δ values for all, one hundred in total, mean values of the examined samples. It should be noted that the samples were also analyzed in order to measure the ratio of sulfur stable isotopes (measured as *δ*^34^S_V-CDT_ (‰)); however, the content of sulfur was 0.00–0.04% *w*/*w*, which was not enough for a reliable calculation of δ values. The results obtained by the analysis are depicted in Figure 2. 

The next stage in this analysis was to find out if it was possible to determine, based on our data, a decision rule to discriminate the samples coming from the PDO zone of “Vostizza”, that is, the Egialia area, against other areas cultivating currants in southern Greece. For this reason, a classification analysis was performed. Classification analysis is a common method used to classify samples based on their chemical composition. It is particularly useful when dealing with complex datasets, where it is difficult to manually classify each sample. Classification analysis typically involves developing a statistical model that uses the measured features or variables (such as the values of δ ratios) to predict the class membership of each sample. The model is typically developed using a training dataset, where the class membership of each sample is already known, and then validated using an independent test dataset. In this study, the classification analysis was based on the “party” R package by using 70% of the data as the training set, and the remaining 30% was used as the testing set for analysis. The following table (Table 2) presents the results of this analysis, and Figure 3 presents the decision tree analysis resulting from the classification analysis. Figure 3 depicts that based on the values of δ^13^C_V-PDB_ (‰), almost three out of four samples can be classified correctly with this approach.

The next step in the analysis was to increase the accuracy of the classification approach. Since our samples were classified into two groups, those coming from the “Vostizza” PDO zone and those coming outside the PDO zone, a binomial logistic regression was applied. Binomial logistic regression is a statistical method used to model the relationship between a binary dependent variable and one or more independent variables known as the predictor variables or covariates. The dependent variable can take only two values (typically zero and one) representing two possible outcomes, such as success/failure, true/false, or yes/no, and in this case, if a sample came from the PDO zone or not. The goal of a binomial logistic regression is to estimate the probability of the dependent variable taking a specific value (usually one), given the values of the independent variables. Since the probability of an event must lie between zero and one, it is impractical to model probabilities with linear regression techniques because a linear regression model allows the dependent variable to take values greater than one or less than zero. This is achieved by fitting a logistic function to the data, which transforms the linear combination of the independent variables into a probability value that ranges from zero to one [37]. 

It was assumed that samples coming from different areas had different *δ* ratios for stable isotopes. These ratios were formulated by different parameters, including soil characteristics, climate conditions, and cultivation techniques, which created a unique set of characteristics that altogether formulated these stable isotope rations [12]. Nevertheless, these parameters could have multiple or multidimensional effects on the final classification. Thus, a given value of a stable isotope ratio might be associated with many characteristics and affect the final classification results. 

Finally, the regression coefficients were estimated through an iterative maximum likelihood method. Table 3 presents the estimation of the model coefficients that can be used to predict if a sample comes from the “Vostizza” PDO zone or not. It should be noted that the coefficients of the estimated model indicate the direction and the strength of the relationship between each independent variable, that is, δ^15^N_AIR_ (‰) and δ^13^C_V-PDB_ (‰), and the dependent variable, which is coming from the PDO zone or not. In addition, the estimated coefficients can be used to make predictions on new data of δ^15^N_AIR_ (‰) and δ^13^C_V-PDB_ (‰) combinations.

Table 4 depicts the classification table resulting after the evaluation of the goodness of fit of the model using the Akaike information criterion (AIC) and the receiver operating characteristic (ROC) curve. These metrics were used to compare different estimated models before choosing the best-fitting one.

The above results indicate that by using the binomial logistic regression approach, 85% of the samples outside the PDO zone and 80% of the samples coming from the PDO zone can be correctly classified. Thus, more than four out of five samples can be correctly classified. The difference of the Corinthian currants species tested might also contribute to their variation in terms of of δ^13^C, as it has been proven that different coffee species varied in δ^13^C [38]. In a similar study, Li et al. [39] showed that carbon isotope ratio could be used to identify the geographical origins of Schisandra fruits in China. The δ^13^C values of Schisandra fruits increased with latitude, which was possibly due to the comprehensive action of multiple factors, such as climate and species. Nevertheless, carbon isotope ratio is not sufficient to discriminate between different origins when the regions are narrow and neighboring, such as those presented in this study.

Thus, the results of this analysis indicate that further stable isotope ratios should be included in the analysis in order to increase the overall accuracy level. A follow-up study will combine trace elements with stable isotope ratios to analyze a larger set of samples with more geographically diverse samples and adulterated samples.

## 4. Conclusions

Stable isotope analysis has been used in many products to authenticate their geographical origins. This is a preliminary approach since there are no previous studies focusing on stable isotope analysis of “Vostizza” PDO currants. This study aimed to investigate if the geographical origin of the most known Greek currant—“Vostizza”—could be differentiated from other currants registered as PGI products from neighboring areas in the same prefecture, namely Peloponnese. Currants are historically important for the economy of both Peloponnese and Greece; thus, determining the origins of these products is essential to establish their authenticity and detect any possible commercial fraud. Beyond the stable isotope ratio analysis and the determination of δ^15^N and δ^13^C, an attempt was made in order to find the best alternative chemometric approach to verify the authenticity of the Greek “Vostizza” currants. It was not possible to detect δ^34^S due to the low sulfur content. This fact made the second stage of the discrimination procedure even more difficult since the analysis had to be based on only two variables, namely δ^15^N_AIR_ (‰) and δ^13^C_V-PDB_ (‰). For the statistical analysis, one-way ANOVA and classification analysis were performed, which are the most common approaches to discriminate geographical origin. 

However, due to the small number of samples and the use of δ values of only two elements, as well as the fact that the examined products came from neighboring areas, the results and, consequently, their explanatory strength were not sufficient. However, applying a different approach, namely binominal logistic regression, managed to improve the explanatory strength and correctly classify 85% of the samples outside the PDO zone and 80% of the samples coming from the PDO zone. Even though these results have been improved, they are still not satisfying to effectively discriminate samples coming from such close neighboring areas. 

The results of the stable isotope analysis suggest that geographical discrimination is not feasible based only on, firstly, carbon isotopes and, secondly, nitrogen isotopes. To achieve sufficient discrimination between different origins, especially when regions are too narrow and neighboring, such as those presented in this study, further research is needed in order to include more samples from the areas producing these products and to analyze more stable isotopes (such as hydrogen and oxygen).

## Figures and Tables

**Figure 1 foods-12-01672-f001:**
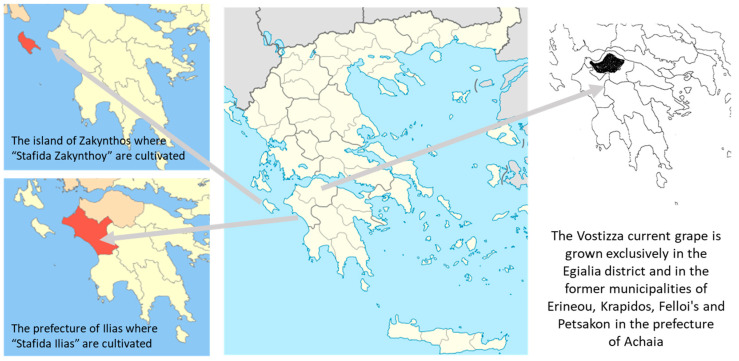
The geographical area where currants registered as a PDO product with the name “Vostizza” are produced. At the left side of the figure, there are the other two areas with registered Corinthian currants.

**Figure 2 foods-12-01672-f002:**
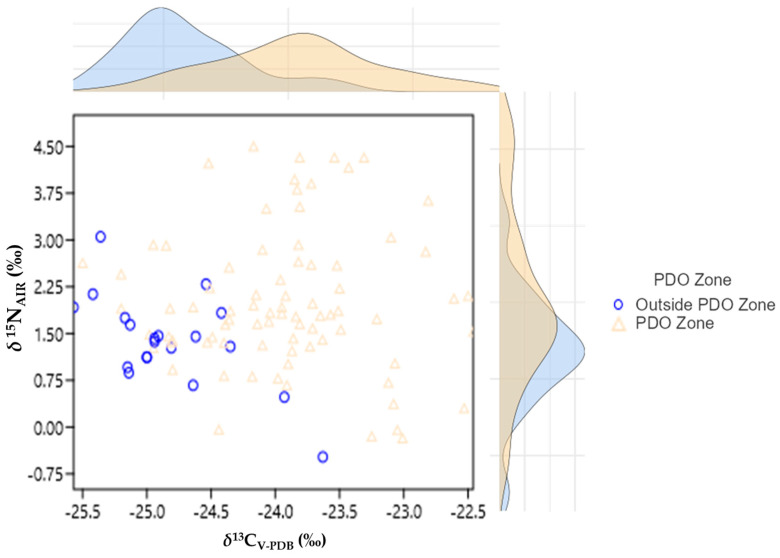
Results for the currant samples’ δ^15^N_AIR_ (‰) and δ^13^C_V-PDB_ (‰) values in a 2D scatterplot with densities.

**Figure 3 foods-12-01672-f003:**
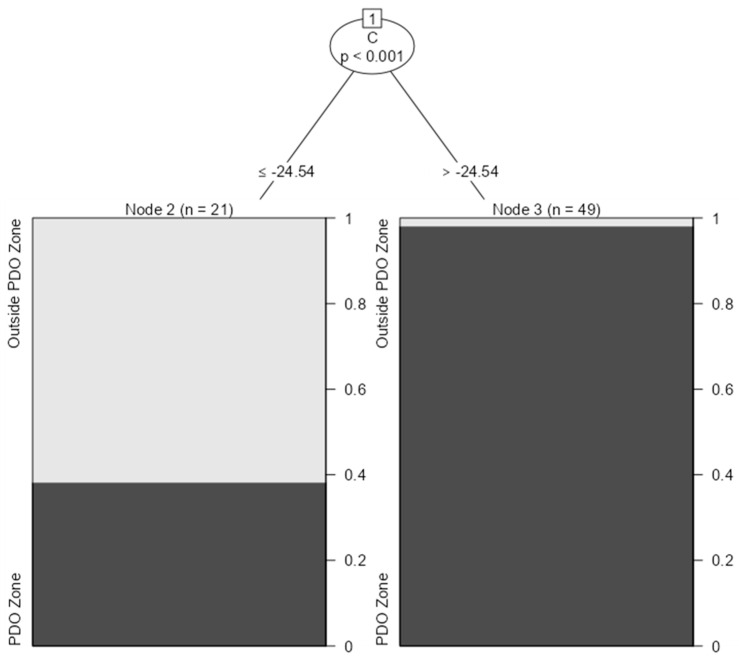
Decision tree analysis to discriminate currant samples coming from the PDO zone of “Vostizza” against samples outside of this zone.

**Table 1 foods-12-01672-t001:** One-way ANOVA results (mean values and standard error).

*δ* (‰)	Area	N	Mean *δ* (‰)	Std. Error	ANOVA (F-Value)	Sig.
*δ*^15^N_AIR_ (‰)	PDO Zone “Vostizza”	80	1.38	1.115	9.05	0.004
	Outside PDO Zone	20	2.01	0.735		
*δ*^13^C_V-PDB_ (‰)	PDO Zone “Vostizza”	80	−23.93	0.668	46.85	<0.001
	Outside PDO Zone	20	−24.83	0.486		

**Table 2 foods-12-01672-t002:** Confusion matrix according to the classification analysis.

Observed	Predicted	
	Outside PDO Zone	PDO Zone
Outside PDO Zone	3	5
PDO Zone	3	19

Note: overall accuracy: 0.733 (95% CI ranges from 0.541 to 0.877).

**Table 3 foods-12-01672-t003:** Binomial logistic regression model and its coefficients.

Predictor	Estimate/Coefficient	SE	Z	*p*
Intercept	77.37	18.508	4.18	<0.001
*δ* ^15^N_AIR_ (‰)	1.57	0.561	2.81	0.005
*δ*^13^C_V-PDB_ (‰)	3.21	0.768	4.18	<0.001

Note 1: The estimates represent the log odds of “area = 1” PDO Zone vs. “area = 0” Outside PDO Zone. Note 2: Model Fit measures: *x*^2^ = 41.3, df = 2, *p* < 0.01, R^2^_McF_ = 0.413, R^2^cs = 0.0339, and R^2^_N_ = 0.535.

**Table 4 foods-12-01672-t004:** Binomial logistic regression: classification analysis.

Observed	Predicted		
	Outside PDO Zone	PDO Zone	% Correct
Outside PDO Zone	17	3	85%
PDO Zone	16	64	80%

Note 1: The cut-off value is set to 0.8. Note 2: Predictive measures: Accuracy = 0.810, Specificity = 0.85, Sensitivity = 0.80, and AUC = 0.909.

## Data Availability

All related data and methods are presented in this paper. Additional inquiries should be addressed to the corresponding author.

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
