# Peer review of "Application of Stable Isotope Analysis for Detecting the Geographical Origin of the Greek Currants “Vostizza”: A Preliminary Study"

_foods, 2023, doi:10.3390/foods12081672_

Round 1
Reviewer 1 Report
#Add to the title: "-a feasibility study"
#Abstract needs to be rewritten. It contains almost no information on the results and interpretation.
Introduction
#rewrite first sentence, unclear.
#Paragraph 43-62 is all about food authenticity. however, your study is NOT about authenticity, it is about geographic origin. Shorten. Add a paragraph about geographic origin.
# Paragraph 63-74: Ad references to all methods listed.
# Row 71-74: rewrite sentence. Your study is NOT about animals, explain parameters for plants in detail.
# Row 74: Sulfur isotopes do NOT depend on soil structure.
#Remove header "The examined product", this is part of the introduction. Alternatively use it as sub-header.
#Paragraph 89-100: Add references to all claims listed.
#Row 115: Describe the PDO region of Vostizza with reference. Does it exactly correlate with the description of your ref. 16?
#Rows 121-125: Add references to all claims.
#Paragraph 129-149: Add references to claims.
Materials and Methods:
# add table of samples with all individual samples
#Figure 1: add the regions of the other two PDO/PGI currants regions in Greece. In the inlay map use a more detailed view to show the Achaia prefecture with the municipalities. Add the localities of all samples in the inlay map (you probably will need to enlarge it significantly, and you can reduce the size of the overview-map.
#Row 175: remove "primarily"
# Explain why you extract the homogenized sample with the acetone/water mixture.
#In methods description add the values for the sulfur isotope standards.
Results
#Add a table of all results of all samples. Add 34S-results that could be measured.
#Figure 1: Use signature that differ in shape that the figure is readable in black-and-white. use same signatures as in Fig. 2.
Discussion
#Rewrite discussion. Currently this is a kind of repetition of the introduction. Interpret, explain the data.
Conclusions
#Rewrite conclusions
References
# replace references 8-11 refering to animal commodities by references dealing with vegetable food.
Generally, I suggest that you measure some 5-10 samples of each of the other two Greek currants PDO/PGI regions, to give the study more relevance. Also, it would be good, if H- and O-isotopes could be measured.
Individual samples must be listed, together with their results.
Author Response
Dear reviewer,
We would like to thank for your valuable comments that can really improve our work.
We have tried to incorporate your suggestions in our manuscript (see the attached file)

Reviewer 2 Report
Dear authors,
The study is a basic application of isotope analysis on a product with, indeed, special commercial value, since it is a PDO product. Considering the fact that stable isotope ratios have been used extensively for origin verification purposes, the study needs to show novelty as a dataset of value. The limitation to only 2 stable isotope ratios (nitrogen and carbon), which happen to be (based on the principles of stable isotope variation in nature) the least significant ones among the five most common isotope ratios (hydrogen, oxygen, carbon, nitrogen and sulfur) in regards to applications of origin verification, makes the dataset less significant. Additionally, the statistical approach seems interesting, but after all, it becomes useless, when considering that there are only 2 parameters in the dataset, only 2 groups (PDO and non-PDO) and only 100 samples. The biplot of N and C isotope signatures reflects all the data in this study.
Some specific points:
Lines 37-38: this is not the definition of authenticity; authenticity does not have to do anything with quality and regulations. Please refer to an official source of definition (eg. GFSI, CEN, or even peer-reviewed articles about food fraud terms).
Lines 43-44: repetition
Lines 47-48: inaccurate; verifying authenticity mitigates risks about safety, it does not help ensuring food safety
Lines 71-74: please rephrase. 'ratio of elements', 'nutrition of animals' referring only to C and N isotopes, do not describe Stable Isotope Analysis.
Lines 83-86: repetition
Section 2 The Examined Product: this is too long, there is no need for such a detailed description. Please reduce
Section 3.2 Sample preparation, Lines 177-180: It is not clear why the authors extract the samples with acetone/water. It is recommended to explain the reason for the chosen sample prep method.
Section 3.3 EA-IRMS analysis: This section needs significant improvement. Since this is the sole method used in this study, and especially seeing that the discussion is based only on N and C isotope ratios, a solid methodological description should be given, with precise analytical information. Please refer to doi.org/10.1515/pac-2021-1108 for guidance on how stable-isotope delta results should be presented. Further, the authors used only 1 reference material for normalizing the isotope data per element. This is not good practice, since there is no slope (stretching) used. Additionally, although currants (C3 plant) and IAEA-600 have similar range of 13C/12C, the QC used (C4-plant) is totally out of the carbon isotope range.
Line 215: 1-2mg of plant material is unlikely to provide enough S content for isotope analysis. Why not using 4-5mg, incl. V2O5 for 34S analysis?
Lines 222-244: PCA is not suitable at all for this dataset. As the authors say themselves, PCA helps to process multi-dimensional datasets. This is a set of 100 samples with 2 parameters.
Lines 249-325: Too complex statistics for this dataset. Figure 1 and Table 1 illustrates the importance of it. We see overlapping N and C isotope signatures between the PDO and the non-PDO samples. There can't be any decision-making index out of this dataset.
Line 319: The scales are very different between the two studies. Carbon isotopes cannot discriminate between different origins, when the regions are narrow and neighboring, such as those presented in this study.
The study needs to be presented only as preliminary, showing some N and C isotope data of currants from the given regions. I agree with line 368, plus the need for more analytical parameters.
After major revision (incl. removal of statistics, highlighting the preliminary status of the dataset, and correcting the description/presentation of the analytical approach), the article might be reconsidered for publication.
Author Response

(The authors gave the same response as above.)

Reviewer 3 Report
I do not think that the PDO and non-PDO samples are sufficiently well separated. This may well be because the non-PDO samples are collected very near to the PDO samples. Very often the boundaries within which the PDO samples are produced are somewhat arbritary.
Author Response
Dear Reviewer,
Thank you for your comment,
Definitely, there is a point in your comment, However the whole work done (even with its limitation) attempts to shed some light in this very interesting subject.
Round 2
Reviewer 1 Report
Add more (Meta-)data to the supplement table.
Rewrite first sentence in Introduction
Rewrite the paragraph about geographic origin and authenticity, these are two totally different things!
Include Supplementary Table 1 in main text. Give a map of the PDO-area with the exact position of each PDO-sample.
Explain the mechanisms behind the d13C- and d15N- parameters. Are they sustainable and reliable?
Author Response
Dear reviewer,
Thank you for your comments and your suggestions
Please find our answers in the attached file.

Reviewer 2 Report
Dear authors,
The revision added mostly some improvement and clarity on the terminology, structure and descriptive text. However, the methodological part, incl. the discussion, is not supported by the results.
In very simple words, Figure 1 (actually, second figure in the manuscript showing the N and C isotope results) reflects 100% the quality of the current study. Statistics are not needed to prove that for authoritative decisions that may lead to legal procedures and strict consequences to businesses, the discrimination of PDO and non-PDO Vostizza cannot be supported by this dataset at all. Please just see the N and C isotope signatures of the PDO and non-PDO samples, they overlap 100%. How could you assess a fingerprint of d15N=+1,5 and d13C=-24,7? PDO or non-PDO? Please put aside the statistical processing in the study, this is not appropriate for a dataset of 2 parameters, from which one us confirmed as useless (nitrogen). And surely PCA is not applied in such datasets (yes, it is applied in authenticity studies, but when the number of parameters is much higher) and it must be removed. In the way that the authors suggest, there could be also 20 more statistical methods applied (and fail).
The manuscript needs, additionally to the Preliminary description, which is already added, to exclude any statements or conclusions regarding successful discrimination of PDO and non-PDO Vostizza. To the eyes of an expert evaluator or industry auditor in food authentication, the presented results actually prove that the stable isotope signature of carbon only is not appropriate for this purpose.
Author Response
Dear reviewer,
Thank you for comments, we have incorporated in this version of the manuscript
We have removed from the text the PCA Analysis, and we have removed based on you comment any statement for sufficient discrimination

Reviewer 3 Report
The corrections are fine
Author Response
Dear Reviewer,
Thank you for comments,